# Patient-Reported Outcome Measures in Patients with and without Non-Expandable Lung Secondary to Malignant Pleural Effusion—A Single-Centre Observational Study

**DOI:** 10.3390/diagnostics14111176

**Published:** 2024-06-03

**Authors:** Jesper Koefod Petersen, Katrine Fjaellegaard, Daniel Bech Rasmussen, Gitte Alstrup, Asbjørn Høegholm, Jatinder Sing Sidhu, Rahul Bhatnagar, Paul Frost Clementsen, Christian B. Laursen, Uffe Bodtger

**Affiliations:** 1Respiratory Research Unit PLUZ, Department of Internal and Respiratory Medicine, Zealand University Hospitals, Roskilde and Næstved, 4000 Roskilde, Denmark; kafj@regionsjaelland.dk (K.F.); dbrs@regionsjaelland.dk (D.B.R.); gahn@regionsjaelland.dk (G.A.); ahoe@regionsjaelland.dk (A.H.); jssd@regionsjaelland.dk (J.S.S.); paul.frost.clementsen@regionh.dk (P.F.C.); ubt@regionsjaelland.dk (U.B.); 2Institute of Regional Health Research, University of Southern Denmark, 5000 Odense, Denmark; 3Department of Respiratory Medicine, Odense University Hospital, 2900 Hellerup, Denmark; rahul.bhatnagar@bristol.ac.uk; 4Academic Respiratory Unit, University of Bristol, Bristol BS8 1TU, UK; 5Odense Respiratory Research Unit (ODIN), Department of Clinical Research, University of Southern Denmark, 5000 Odense, Denmark; 6Copenhagen Academy for Medical Education and Simulation (CAMES), Rigshospitalet, University of Copenhagen and the Capital Region of Denmark, 2100 Copenhagen, Denmark

**Keywords:** non-expandable lung, malignant pleural effusion, patient-reported outcomes, survival

## Abstract

Background: Malignant pleural effusion (MPE) affects up to 15% of patients with malignancy, and the prevalence is increasing. Non-expandable lung (NEL) complicates MPE in up to 30% of cases. However, it is not known if patients with malignant pleural effusion and NEL are more symptomatic in activities of daily living compared to patients with MPE with expandable lung. Methods: This was an observational study on consecutively recruited patients with MPE from our pleural clinic. Before thoracentesis, patients completed patient-reported outcomes on cancer symptoms (ESAS), health-related quality of life (5Q-5D-5L), and dyspnoea scores. Following thoracentesis, patients scored dyspnoea relief and symptoms during thoracentesis. Data on focused lung ultrasound and pleural effusion biochemistry were collected. The non-expandable lung diagnosis was made by pleural experts based on radiological and clinical information. Results: We recruited 43 patients, including 12 with NEL (28%). The NEL cohort resembled those from previous studies concerning ultrasonography, pleural fluid biochemistry, and fewer cases with high volume thoracentesis. Patients with and without NEL were comparable concerning baseline demography. The 5Q-5D-5L utility scores were 0.836 (0.691–0.906) and 0.806 (0.409–0.866), respectively, for patients with and without NEL. We observed no between-group differences in symptom burden or health-related quality of life. Conclusion: While the presence of NEL affects the clinical management of recurrent MPE, the presence of NEL seems not to affect patients’ overall symptom burden in patients with MPE.

## 1. Introduction

Up to 15% of all cancer patients develop malignant pleural effusion (MPE) during the course of their disease [1,2,3]. The number of patients diagnosed with and surviving malignancy is increasing, and thus the incidence of MPE is increasing too [4]. MPE is a serious condition as it denotes metastatic disease in almost all cases. Furthermore, MPE can be complicated by the inability of the lung to fully expand (non-expandable lung (NEL)), with the inability of the pleural surfaces to attain apposition [5,6]. NEL is caused by bronchial tumour obstruction, chronic atelectasis, or reduced elasticity of the visceral pleura, and affects up to 30% of patients with MPE [6,7,8,9,10]. There is no available treatment for NEL in the vast majority of patients, so therapeutic management focuses on the palliation of the MPE and underlying disease [3,6,11,12].

MPE is associated with a significant symptom burden [13,14]. Compared to MPE with expandable lung, MPE with NEL is clinically associated with lower pleural effusion volumes at thoracentesis, thoracentesis-related pain, and cough, and biochemically with transudative rather than exudative effusions [15,16,17]. In patients with pleural mesothelioma, NEL is found to be associated with increased symptom burden and poorer survival [18]. These findings may not be extrapolated to non-mesothelioma malignancies, where the presence of MPE—in contrast to mesothelioma—by definition denotes disseminated disease (stage IV) [19]. Therefore, we aimed to investigate whether NEL confers additional symptoms to those related to disseminated malignancy in the presence of MPE [20].

## 2. Materials and Methods

A prospective single-centre study was performed at a respiratory clinic with a dedicated outpatient pleural service (Pleural Clinic, Department of Internal and Respiratory Medicine, Zealand University Hospital, Naestved, Denmark). Patients were eligible if >18 years, had MPE, and needed a thoracentesis due to an effusion larger than 2 cm (ultrasound measurement). Exclusion criteria were life expectancy shorter than 3 months, inability to understand written or spoken Danish, pregnancy, and contradictions to therapeutic thoracentesis.

Following inclusion, baseline data and patient-reported outcome measures (PROMs) were collected, followed by thoracic ultrasound (TUS), TUS-guided thoracentesis, and then post-thoracentesis TUS and PROMs.

### 2.1. Non-Expandable Lung (NEL) Definition

As initially proposed by Lan et al. [8] and refined in the study by Salamonsen et al. [21], we defined NEL based on a combination of clinical observations and post-thoracentesis radiology (Appendix A). Two independent interventional pulmonologists subspecialised in pleural disease (AH and JSS) assessed study data after four months of follow-up and classified cases as non-expandable lung (NEL), expandable lung (EL), or “Unable to determine”. In the event of disagreement between assessors, a third subspecialised interventional pulmonologist (UB) assessed study information and consensus was achieved. The assessors were blinded to the decision of the fellow assessor, and were provided with underlying diagnoses, relevant clinical observations during thoracentesis (e.g., chest tightness or severe coughing alleviated or not by thoracentesis pause, air in the chest tube, and undrainable pleural effusion despite unblocked chest tube), and available radiology reports and images (before and after thoracentesis) as well as performed PROMs.

### 2.2. Baseline Data

Basic demographics, including sex, smoking history, comorbidity, existing radiology reports and images, blood and pleural biochemistry (protein, albumin, and lactate dehydrogenase (LDH)) where available, and final cause of pleural effusion were collected at study inclusion and at the four-month follow-up. Survival status was registered at the four-month follow-up where a final diagnosis was made, and again twelve months later.

### 2.3. Patient-Reported Outcome Measures (PROMs)

PROMs were screened for validation on patients with NEL and in Danish but could only be found for the latter. We identified PROMs that were well studied in patients with malignancy (or dyspnoea with minimal important differences specified in patient groups with malignancy) and validated in Danish.

#### 2.3.1. Pre-Thoracentesis PROMs

PROMs were completed by the patient after initial study inclusion was completed and while sitting ready for planned thoracentesis.

EuroQol EQ-5D-5L measures health-related quality of life [22]. The descriptive system comprises five dimensions (mobility, self-care, usual activities, pain/discomfort, and anxiety/depression) scored by the patient on a 5-level Likert scale (no/slight/moderate/severe/extreme problems). Responses were converted to a utility score utilising Danish national validated scores [23], with 1 being least possible symptoms. It also contains a 100 VAS score on self-rated health, with higher scores indicating better quality of life.

The Edmonton Symptom Assessment Score (ESAS) measures cancer-related disease burden [24]. It comprises nine predefined symptoms and one patient-defined symptom scored by the patient on an 11-point Likert scale (0 = no symptoms, 10 = worst symptoms imaginable). Scores were dichotomized into “none-to-mild” (score 0–3) or “moderate-to-severe” (score 4–10) [24,25,26].

The Medical Research Council dyspnoea scale (modified version) (mMRC) measures dyspnoea on a 5-point Likert scale completed by the patient (0 = I only get breathless with strenuous exercise, 4 = I am too breathless to leave the house, or I am breathless when dressing/undressing). Additionally, the ESAS dyspnoea symptom score was included to ascertain dyspnoea burden.

The Borg Rating of Perceived Exertion (MBS) scale (modified) measures the severity of perceived exertion on an 11-point Likert scale completed by the patient (0 = no effort at all, 10 = very, very hard). Scores were dichotomized as mild–moderate (score 0–3) or severe (score 4–10 ) [27].

#### 2.3.2. Post-Thoracentesis PROMs

Following thoracentesis and immediately prior to leaving the outpatient clinic (having dressed), patients were asked to describe the following symptoms:

Symptoms during thoracentesis, evaluating experienced chest tightness or pain relating to thoracentesis on a 4-point Likert scale (no/slight/moderate/marked symptoms) completed by the patient.

Symptom change in dyspnoea and in wellbeing after thoracentesis, each evaluated on a 6-point Likert scale (“markedly better”, “slightly to moderately better”, “unchanged”, “slightly worse”, “moderately worse”, and “markedly worse”) completed by the patient. “Markedly better” and “slightly-to-moderately better” defined a “clinical benefit” in the present study.

### 2.4. Thoracentesis and TUS-Protocol

Two physicians (JKP and KF) performed all procedures. Both were trained and certified in TUS and thoracentesis according to national guidelines of the Danish Respiratory Society [28]. Ultrasound assessments were performed using LOGIQ S8 (GE Healthcare, Wauwatosa, USA) using the C1-6-D curved (2–5 Mhz) transducer in abdominal preset. Thoracentesis was performed with both a diagnostic and therapeutic intent, aiming for optimal patient symptom relief. Thoracentesis was aborted when more than 2 L of pleural fluid was accumulated or patient symptoms of severe cough, chest pain, or general discomfort became clinically significant.

TUS was performed before and after thoracentesis. TUS initially assessed both hemithoraces with measurement performed on the area of primary effusion and adjacent lung zones where thoracentesis was planned. A full ultrasound protocol has been previously published [29].

Large-volume thoracentesis was defined as more than 760 mL of pleural fluid being removed during thoracentesis as per the study by Mishra and colleagues [30].

Pre-thoracentesis TUS assessment [31] included the evaluation of visible atelectasis (yes/no) pleural, parietal pleural thickening over 1 cm (yes/no), septations (yes/no), lung sliding apically of the effusion (yes/no), and swirling sign(31) (yes/no). 

Post-thoracentesis TUS assessment: Following thoracentesis, the hemithorax was assessed using ultrasound for complete drainage status (less than 0.1 L fluid assessed by visual estimation) and pleural apposition.

### 2.5. Follow-Up Data

All patients had a chest X-ray (CXR) performed after thoracentesis, except when a same-day chest computer tomography (CT) was booked by the treating oncologist or physician. 

Final diagnosis was recorded 4 months from thoracentesis using the electronic patient records. 

## 3. Results

Between May 2021 and December 2021, 132 patients were screened at 289 visits, and 49 patients were included in the study (see flowchart, Figure 1). Six patients with a final diagnosis of non-malignant disease were excluded, and one patient (NEL) was unable to complete all questionnaires due to fatigue. Thus, the study included 43 patients with complete PROMs data from 42 patients.

A total of 12/43 patients were diagnosed with NEL (29%). Inter-rater agreement on probable/definite NEL vs. probable/definite EL was “very good” with a Kappa of 0.93 (SE 0.16), and with an overall weighted Kappa of 0.79 (SE 0.13) on the non-dichotomised assessments. Table 1 shows basic demographics stratified by NEL status.

### 3.1. Primary Outcome

Dyspnoea before thoracentesis did not consistently differ between groups regardless of PROMs used (Table 2). Furthermore, no inter-group difference in dyspnoea relief following thoracentesis was observed (*p* = 0.14). In the subgroup with large thoracentesis volumes (predefined as >760 mL), patients with EL reported more dyspnoea relief than patients with NEL (*p* = 0.047).

### 3.2. Secondary Outcomes

Neither overall symptom burden (Table 3) nor health-related QoL (Figure 2) was significantly worse in patients with NEL compared to EL. EQ-5D-5L median utility scores showed no between-group difference (0.836 & 0.806).

Biochemical and ultrasound findings are listed in Table 4.

All patients with NEL had undergone a previous thoracentesis, compared to 74% of patients with EL (*p* = 0.043). Effusions in the NEL group were characterised by higher levels of LDH (360 U/L vs. 165 U/L, *p* = 0.007) and a higher ratio between pleural fluid and plasma LDH (7.3 U/L vs. 1.2 U/L, *p* = 0.015). Thoracic ultrasound in the NEL group more often demonstrated pleural septations and rarely showed lung sliding. Median survival from first thoracentesis was numerically lower in the NEL group but the difference did not reach significance (NEL: 76 (IQ range 59–111) vs. EL: 116 (IQ range 59–287) days, *p* = 0.23). For overall survival from the diagnosis of malignancy, the Kaplan–Meier survival curve (Figure 3) showed a significant between-group difference (*p* = 0.024)

## 4. Discussion

This is the first prospective study to describe symptom burden in patients with NEL in the context of MPE in comparison to those without NEL. We found no differences in dyspnoea or symptom burden between patients with and without NEL despite using a comprehensive collection of recognised PROMs concerning malignancy or pleural effusions. The NEL group displayed typical characteristics shown in earlier studies such as lower pleural volume removed [32], lower symptoms relief with large thoracentesis volumes [30], and a greater degree of incompletely drained effusions, as well as lower levels of pleural fluid albumin and protein, and elevated lactate dehydrogenase (LDH) [32]. This would support our NEL cohort being comparable to previous studies [32,33]. We expected a higher degree of baseline symptom burden in patients with NEL and MPE, as these patients—in addition to diaphragm immobility induced by pleural effusion—also suffer from impaired lung movement due to NEL [34].

We investigated if living with NEL is associated with a greater subjective symptom burden before having a planned thoracentesis. EQ-5D-5L utility scores were calculated (high scores equal low symptom burden), and compared to national reference values. Scores were calculated from a baseline of 1, with each trajectory subtracting a standard reference value according to the severity of the given trajectory. EQ-5D-5L median utility scores (0.836 and 0.806) did not differ significantly and reflect the VAS scores from the questionnaire. For reference, Danish reference values have been calculated at 0.90 [35]. Our median scores are markedly higher than those found in a comparable population with MPE (0.55–0.71) from the UK [36]. This may reflect cultural differences or more likely that the basic reference values (used to adjust the domains) may be markedly different—these are currently under re-evaluation [37].

Though our study seems to suggest that NEL in MPE is not associated with an overall excess of symptoms, the presence of NEL in MPE poses a variety of challenges to the clinician. This includes confusion with iatrogenic pneumothorax after drainage, the drainage of lower effusion volumes due to patient discomfort, the lower efficacy of large thoracentesis volumes, the lower efficacy of talc pleurodesis, and the delayed insertion of indwelling pleural catheter (IPC). All these are likely to affect patients’ quality of life during MPE management. Thus, the correct and early identification of NEL is crucial for optimal MPE management.

Our study is one of the few prospective studies focusing on patients with MPE and stratified by NEL status. We found that about half of the patients experience symptom relief following thoracentesis. Psallidas et al. [38] found that 71% (10/14) of patients with NEL experienced dyspnoea relief following thoracentesis versus 89% (39/44) of patients with EL. Possible differences in NEL definition or population may account for the observed effect difference, but neither NEL definition or descriptive data on the NEL population is available in that paper [38].

NEL has no “gold standard” diagnostic test and is usually recognized from patient symptoms and radiological findings. It has been suggested that pleural manometry could play a diagnostic role, but studies have shown significant differences in initial pleural pressure and elastance in patients with and without NEL [8,39].

Our study has several strengths and weaknesses. Firstly, we consider it as a strength to assess the overall symptom burden associated with malignancy and impaired health, as this acknowledges the huge variability in symptoms imposed by cancer [40]. Secondly, the consecutive inclusion of a broad variety of patients with suspected or known MPE is a strength as our patients represent the heterogeneous population living with MPE. Thirdly, PROMs being answered just before thoracentesis should represent the potentially most symptomatic phase, without confusing responses with thoracentesis-related discomforts. Further, most patients were familiar with the procedure, so bias due to pre-procedure anxiety is expected to be low. Fourthly, our NEL cohort presented several of the known characteristics of NEL known from previous studies, and thus we consider our cohort to be representative. Lastly, the majority of patients (33/43) had a volume drained exceeding 760 mL, which Mishra et al. identified as the mean volume to induce a minimal important difference in dyspnoea measured with a visual analogue scale [30].

The major limitation is the sample size, which hampers firm conclusions. We were unable to perform a sample size calculation due to the lack of previous studies on dyspnoea relief specifically in MPE-related NEL. Despite the lack of a proper sample size calculation, our cohort was large enough to show significant between-group differences concerning biochemistry and ultrasonography, and thus is likely to be also large enough to detect a signal on major differences in symptom burden between NEL and EL. Between-group differences were limited, but it should be noted that there was a numeric difference in the NEL subgroup with metastatic disease that did not achieve statistical significance (*p* = 0.77) We excluded patients with very short life expectancy to avoid confusing patient-reported outcomes with performance, symptoms, and challenges related to overwhelming cancer burden and imminent end-of-life palliative needs. This may have led to selection bias, yet we are unaware if NEL is markedly more prevalent in patients with a very short life span. Likewise, we did not collect data on time living with malignant pleural effusion before study inclusion, so our data on survival may be influenced by lead-time bias. However, the insignificant yet increasing gap in survival curves between NEL and EL (Figure 3) suggests that NEL may represent a later phase in the MPE trajectory.

Several dyspnoea scales exist but no single tool encompasses all dimensions of the complexity of dyspnoea [30,41]. No scale on MPE-related dyspnoea has been linguistically validated in Danish, and therefore several scales were used to measure and compare dyspnoea. Our negative results may thus be due to an inherent inability to measure MPE-related dyspnoea in our cohort via the chosen PROMs. This highlights the need for domestic translation-validated dyspnoea scales for patients with MPE to assess dyspnoea either as a unidimensional VAS-scale [30] or a multidimensional scale such as the Cancer Dyspnoea Scale [42]. 

Finally, there is currently no consensus on optimal time point for the measurement of change in dyspnoea following thoracentesis. In some studies, measurement is performed immediately following thoracentesis while others find that optimal time at two days post thoracentesis [12,34]. We chose to use same-day time points, as we were worried that the ongoing COVID pandemic would impair our data collection if a time point 2 days later were chosen. We consider it unlikely that our patients were in phase with few symptoms at the time of study inclusion, since patients book an appointment for out-patient therapeutic thoracentesis within 1–2 days in our unit, when the patients experience significant discomfort due to recurrent pleural effusion.

The Kaplan–Meier survival graph showed a significant difference between groups, with worse survival in the NEL group. As symptoms between groups are comparable, we suggest that NEL may especially affect overall survival due to lessened respiratory physiological reserves and advanced dissemination, making patients more vulnerable to infection and further disease dissemination. Shunting in the pulmonary arterial system or pulmonary arterial hypertension may play pathophysiological roles, but this needs further scientific assessment.

## 5. Conclusions

The overall symptom burden (including dyspnoea) in patients with MPE is high but appears not to be higher in patients with non-expandable lung. The Kaplan–Meier survival graph suggests that NEL negatively affects overall survival.

## Figures and Tables

**Figure 1 diagnostics-14-01176-f001:**
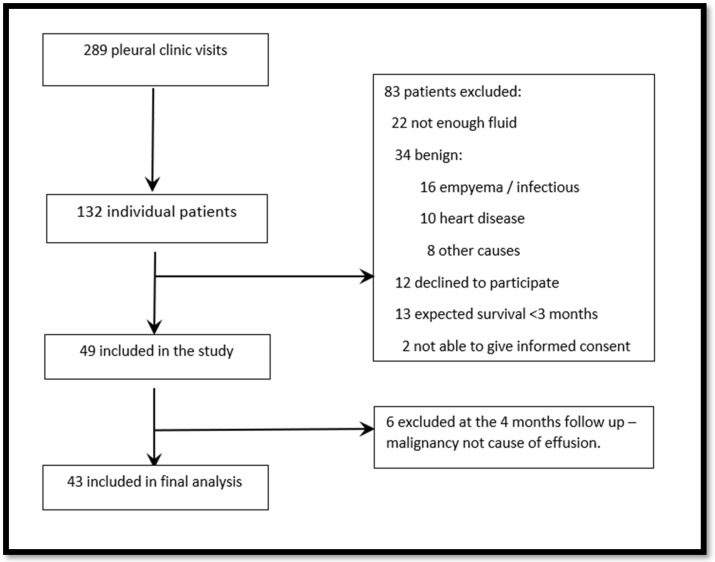
Patient inclusion.

**Figure 2 diagnostics-14-01176-f002:**
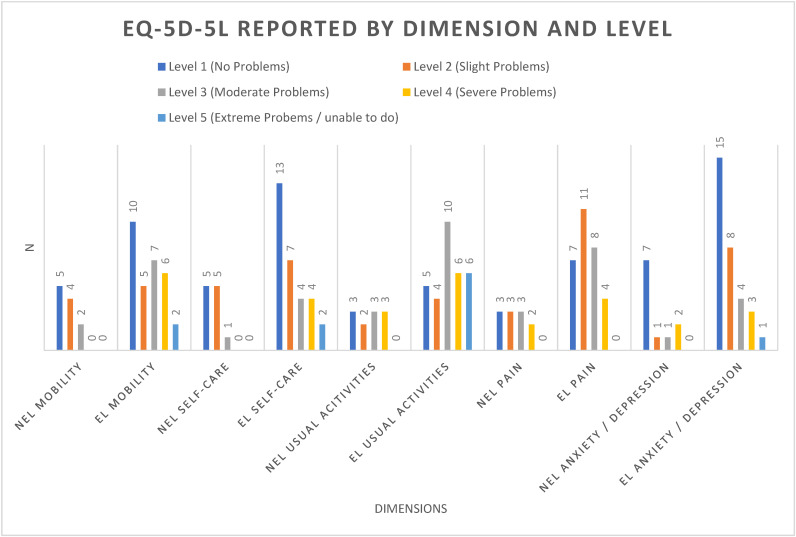
EQ-5D-5L reported by dimension and level.

**Figure 3 diagnostics-14-01176-f003:**
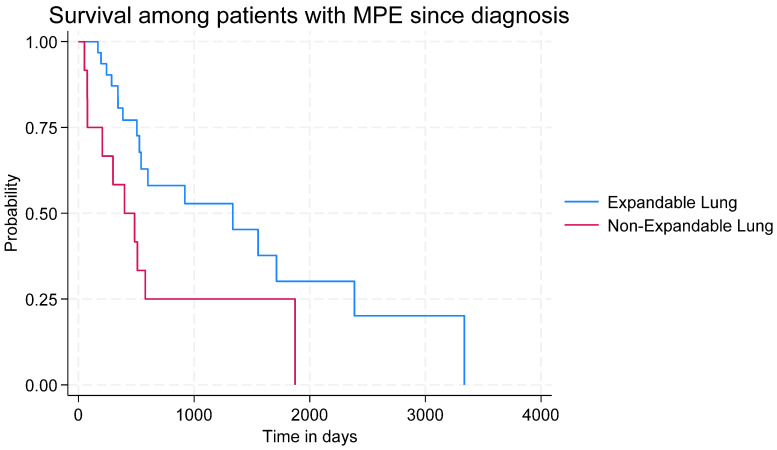
Kaplan–Meier survival graph.

**Table 1 diagnostics-14-01176-t001:** Demographic and key clinical variables.

	Non-Expandable Lung NEL (*n* = 12)	Expandable Lung EL (*n* = 31)	*p*-Value
Age, mean (SD)	75 (8)	71 (8)	0.11 *
Male, *n* (%)	5 (42%)	17 (55%)	0.44 ^#^
Smoking status			0.15 ^#^
Current, *n* (%)	3 (25%)	5 (16%)	
Former, *n* (%)	9 (75%)	18 (58%)	
Never, *n* (%)	0 (0%)	8 (26%)	
Tobacco packyears,	32 (16)	29 (17)	0.62 *
Cancer diagnosis			0.077 ^#^
Lung, *n* (%)	6 (50%)	14 (45%)	
Mesothelioma, *n* (%)	4 (33%)	0 (0%)	
Breast, *n* (%)	2 (17%)	8 (26%)	
Kidney, *n* (%)	0 (0%)	2 (6%)	
Female reproductive, *n* (%)	0 (0%)	2 (6%)	
Melanoma, *n* (%)	0 (0%)	1 (3%)	
Prostate, *n* (%)	0 (0%)	1 (3%)	
Lymph, *n* (%)	0 (0%)	1 (3%)	
Other, *n* (%)	0 (0%)	2 (6%)	
Lung cancers			0.37 ^#^
Adenocarcinoma, *n* (%)	4 (67%)	11 (85%)	
Squamous-cell carcinoma, *n* (%)	1 (17%)	1 (8%)	
Small-cell lung cancer, *n* (%)	0 (0%)	1 (8%)	
Other, *n* (%)	1 (17%)	0 (0%)	
Heart failure, *n* (%)	0 (0%)	3 (10%)	0.26 ^#^
Liver failure, *n* (%)	0 (0%)	0 (0%)	-
Kidney failure, *n* (%)	0 (0%)	1 (3%)	0.53 ^#^
Tuberculosis, *n* (%)	0 (0%)	0 (0%)	-
Performance status ≥2, *n* (%)	5 (42%)	14 (45%)	0.84 ^#^
Preceding thoracenteses, median *n* (range)	3 (2–7)	2 (1–5)	0.28 ^¤^
Time since last thoracentesis, median days (range)	42 (7–81)	22 (7–41)	0.35 ^¤^
Pleural fluid volume drained, mean mL (SD)	990 (621)	1283 (515)	0.13 *
Complete drainage at TUS, *n* (%)	2 (18%)	19 (63%)	0.010 ^#^
Death before follow-up, *n* (%)	5 (42%)	11 (35%)	0.71 ^#^
Survival from first thoracentesis, median days (range)	76 (59–111)	116 (59–287)	0.23 ^¤^
Time from cancer diagnosis to first thoracentesis, median days (range)	3 (−3–236)	84 (0–743)	0.049 ^¤^
Death at 12 months past follow-up, *n*(%)	10 (83%)	17 (55%)	0.083 ^#^
Survival from cancer diagnosis Median days (range)	348 (78–510)	527 (341–1335)	0.11 ^¤^

* Two-sample *t*-test; ^#^ Pearson Chi-square; ^¤^ Wilcoxon rank-sum test.

**Table 2 diagnostics-14-01176-t002:** Baseline dyspnoea scores and change in dyspnoea and wellbeing scores after thoracentesis.

	Non-Expandable Lung *n* =12	Expandable Lung *n* =31	*p*-Value
Post-thoracentesis			
Change in dyspnoea			0.14
Much better	4 (36%)	7 (23%)	
Somewhat better	1 (9%)	11 (35%)	
Unchanged	4 (36%)	12 (39%)	
Somewhat worse	1 (9%)	1 (3%)	
Moderately worse	1 (9%)	0 (0%)	
Wellbeing after thoracentesis			0.25
Much better	3 (27%)	3 (10%)	
Somewhat better	2 (18%)	7 (23%)	
Unchanged	5 (45%)	20 (65%)	
Somewhat worse	0 (0%)	1 (3%)	
Much worse	1 (9%)	0 (0%)	
Pain in or tightness of chest during thoracentesis			0.83
None	8 (73%)	24 (77%)	
Light	2 (18%)	3 (10%)	
Moderate	1 (9%)	2 (6%)	
Severe	0 (0%)	2 (6%)	
Benefit of large volume drainage			0.047
Much better	2 (29%)	7 (26%)	
Somewhat better	0 (0%)	9 (33%)	
Unchanged	3 (43%)	11 (41%)	
Somewhat worse	1 (14%)	0 (0%)	
Moderately worse	1 (14%)	0 (0%)	
Pre-thoracentesis			
MRC			0.80
Breathless only with strenuous exercise	2 (18%)	2 (6%)	
Short of breath when hurrying on level ground or up a slight hill	2 (18%)	6 (19%)	
Slower than most people of the same age on a level surface or have to stop when walking at my own pace on the level	0 (0%)	1 (3%)	
Stop for breath walking 100 m or after walking a few minutes at my own pace on level ground	2 (18%)	5 (16%)	
Too breathless to leave the house	5 (45%)	17 (55%)	
MBS at rest			0.26
Mild-to-moderate (0.5–3)	11 (92%)	25 (74%)	
Severe (>4)	1 (8%)	8 (26%)	
MBS in activity			1.00
Mild-to-moderate (0.5–3)	2 (17%)	5 (16%)	
Severe (>4)	10 (83%)	26 (84%)	
Shortness of breath (ESAS)	5 (2–8)	6 (4–8)	0.41

Large volume drainage pre-defined as >760 mL [30].

**Table 3 diagnostics-14-01176-t003:** Prevalence of moderate-to-severe symptom burden as assessed with ESAS (score 4–10) or EQ-5D-5L (score moderate or worse).

	Non-Expandable Lung *n* = 12	Expandable Lung *n* = 31	*p*-Value
ESAS	*n* = 12	*n* = 31	
Pain, *n* (%)	3 (25%)	12 (39%)	0.49
Tiredness, *n* (%)	6 (50%)	21 (68%)	0.31
Nausea, *n* (%)	1 (8%)	8 (26%)	0.40
Drowsiness, *n* (%)	4 (33%)	15 (48%)	0.50
Appetite, *n* (%)	4 (33%)	19 (61%)	0.17
Dyspnoea, *n* (%)	9 (75%)	25 (81%)	0.69
Depression, *n* (%)	4 (33%)	13 (42%)	0.73
Anxiety, *n* (%)	4 (33%)	13 (45%)	0.73
Wellbeing, *n* (%)	5 (42%)	21 (68%)	0.17
EQ-5D-5L			
Mobility, *n* (%)	3 (23%)	18 (53%)	0.065
Self-care, *n* (%)	1 (8%)	12 (35%)	0.058
Usual Activities, *n* (%)	8 (62%)	25 (71%)	0.51
Pain/discomfort, *n* (%)	7 (54%)	14 (41%)	0.43
Anxiety/depression, *n* (%)	3 (23%)	9 (26%)	0.85
Health, mean VAS score (SD)	64 (28)	58 (26)	0.49
Combined utility score (IQR)	0.836 (0.691–0.906)	0.806 (0.409–0.866)	0.37

**Table 4 diagnostics-14-01176-t004:** Pleural fluid biochemistry (and blood/pleural fluid ratios), and pre-thoracentesis ultrasonographic findings.

	Non-Expandable Lung *n* =12	Expandable Lung *n* =31	*p*-Value
Biochemistry			
Pleural fluid protein, g/L	34.6 (11.5)	40.7 (10.9)	0.22
Pleural fluid albumin, g/L	17.1 (9.5)	23.1 (6.2)	0.058
Plasma albumin, g/L	31.0 (8.1)	28.5 (4.4)	0.30
Albumin gradient, g/L	10.8 (2.3)	6.3 (5.8)	0.080
Pleural fluid LDH, U/L	360 (284–1724)	165 (121–225)	0.007
Serum LDH, U/L	189 (180–190)	200 (170–270)	0.16
LDH gradient, U/L	7.3 (8.6)	1.2 (1.3)	0.015
Pre-thoracentesis ultrasound			
Pleural fluid swirling, *n* (%)	3 (25%)	18 (58%)	0.052
Pleural fluid septations, *n* (%)	6 (50%)	3 (10%)	0.004
Pleural thickening, *n* (%)	10 (83%)	20 (65%)	0.23
Pleural nodules, *n* (%)	3 (25%)	8 (26%)	0.96
Atelectasis/consolidation, *n* (%)	7 (58%)	16 (52%)	0.21
Lung sliding, *n* (%)	2 (18%)	20 (65%)	0.008
Post-thoracentesis ultrasound			
Apposition of pleural lines, *n* (%)	1 (9%)	22 (73%)	<0.001
US assessed full drainage, *n* (%)	2 (18%)	19 (63%)	0.010

Data are presented as mean (SD) or median (IQR). LDH; Lactate Dehydrogenase. Gradient (blood concentration minus pleural fluid concentration).

## Data Availability

All data from the study are presented in this manuscript or in the online Appendix A. For questions concerning data, please contact the corresponding author.

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
