# Peer review of "Patient-Reported Outcome Measures in Patients with and without Non-Expandable Lung Secondary to Malignant Pleural Effusion—A Single-Centre Observational Study"

_diagnostics, 2024, doi:10.3390/diagnostics14111176_

Round 1

Reviewer 1 Report

Comments and Suggestions for Authors

Thank you for the possibility to review the manuscript titled: “Patient-reported outcome measures in patients with and without non-expandable lung secondary to malignant pleural effusion – a single centre observational study”. This is a very interesting manuscript that has practical application in the day-to-day practice. The manuscript provides an overview of 41 articles and cites most of the important literature. The material and methods section deserve a separate appraisal as they are well organized and the design of the study is described explicitly. There are no major objections. There are several minor recommendations that I believe may improve the overall quality of the manuscript.

The “cancer diagnosis” groups from table 1 seem to be identical with a p value of 0.077. However, it may be that there are not enough included cases. There were more cases of metastatic disease in this subgroup and this requires an analysis or inclusion in the limitation section.

Kaplan-Meier survival graphs suggests that NEL negatively affects overall survival. This should be discussed in more details in the discussion section. Please include some ideas on the possible pathogenesis and how non-expanded lung affects the respiratory function in the patient leading to an increased mortality rate. There are important practical implications and prognosis for this group of patients.

Please take into account the recommendations in the spirit of improving the quality of the submission.

Author Response

Thank you for the sound comments, we are keen to present a publication of as high quality possible and value your suggestions.

  • The “cancer diagnosis” groups from table 1 seem to be identical with a p value of 0.077. However, it may be that there are not enough included cases. There were more cases of metastatic disease in this subgroup and this requires an analysis or inclusion in the limitation section.

  • Yes, the group is limited in size and therefore we have added this to the limitation, as an analysis my potentially be underpowered and present risk of both type 1 or type 2 errors. The Discussion contains a section (page 10) on limitations imposed by the restricted sample size and a single centense has been further added to the discussion - line 266.

  • Kaplan-Meier survival graphs suggests that NEL negatively affects overall survival. This should be discussed in more details in the discussion section. Please include some ideas on the possible pathogenesis and how non-expanded lung affects the respiratory function in the patient leading to an increased mortality rate. There are important practical implications and prognosis for this group of patients.
    • Yes, this is indeed interesting, and we have inserted a section in the Discussion as suggested. However, these are post hoc analyses, so any result is at the best hypothesis-generating and not conclusive. Therefore we have tried not to put too much emphasis on possible causes. As the study shows there is no difference in general dyspnea or other measured PROMs, so our suggestions is that the pathogenesis may be related to the advanced dissemination of the malignancy and an end-stage of the disease. The non-expandable lung may increase the risk of pulmonary shunting and may represent a risk for increased pulmonary arterial hypertension. The physiological reserves may be very limted and minor complications such as infection may represent an even greater risk for the patients. This of course may further warrant the involvement of advanced palliative care.

Reviewer 2 Report

Comments and Suggestions for Authors

Dear Authors,

congratulations on your project. Here are my comments.

The introduction is concise and clear. It is well supported by the literature. The focus of the study is well highlighted. The materials and methods are presented effectively. I have no improvements to request in this regard.

The results are well supported by figures and tables. The text is sufficiently easy to read. The discussion extensively analyzes the strengths and limitations of this study. It compares previous studies with the present results and offers an interesting point of reflection. The bibliography is extensive.

In line 66, please specify the meaning of PROMs before using the acronym alone.

Figure 1 could be of better quality.

In line 162, why is the proportion 12 out of 42? Shouldn't the denominator be 43?

Table 1 and Table 2 could have a more consistent style.

Author Response

Thank you for the kind words. We value the sound comments and feedback, we are keen to present a publication of as high quality possible and value your suggestions.

In line 66, please specify the meaning of PROMs before using the acronym alone.

  • Added

Figure 1 could be of better quality.

  • The figure has been updated with focus on a more stringent appearance.

In line 162, why is the proportion 12 out of 42? Shouldn't the denominator be 43?

  • Thank you very much for finding that, even after several readthroughs sometimes vital numbers go unnoticed. Now it’s changed

Table 1 and Table 2 could have a more consistent style.

  • Yes, we note there has been some layout changes to the tables. We have worked to make these more consistent in style as to improve the visual experience of the publication.